# Review of Active Extracorporeal Medical Devices to Counteract Freezing of Gait in Patients with Parkinson Disease

**DOI:** 10.3390/healthcare10060976

**Published:** 2022-05-24

**Authors:** Mónica Huerta, Boris Barzallo, Catalina Punin, Andrea Garcia-Cedeño, Roger Clotet

**Affiliations:** 1Universidad Politécnica Salesiana, Cuenca 010105, Ecuador; bbarzallo@ups.edu.ec (B.B.); bpunin@ups.edu.ec (C.P.); andregarciace@gmail.com (A.G.-C.); 2GRID Research Group, Universidad Internacional de Valencia, 46002 Valencia, Spain; roger.clotet@campusviu.es

**Keywords:** Parkinson Disease, gait recognition, smart device, non-invasive devices, stimulation, freezing of gait

## Abstract

Parkinson Disease (PD) primarily affects older adults. It is the second-most common neurodegenerative disease after Alzheimer’s disease. Currently, more than 10 million people suffer from PD, and this number is expected to grow, considering the increasing global longevity. Freezing of Gait (FoG) is a symptom present in approximately 80% of advanced-stage PD’s patients. FoG episodes alter the continuity of gait, and may be the cause of falls that can lead to injuries and even death. The recent advances in the development of hardware and software systems for the monitoring, stimulus, or rehabilitation of patients with FoG has been of great interest to researchers because detection and minimization of the duration of FoG events is an important factor in improving the quality of life. This article presents a review of the research on non-invasive medical devices for FoG, focusing on the acquisition, processing, and stimulation approaches used.

## 1. Introduction

Parkinson Disease (PD) is the second-most common neurodegenerative disease, after Alzheimer’s disease, among the elderly. At present, there are 10 million people patients with PD worldwide [1,2]. Statistics show that the prevalence of PD is higher in Europe, North America, and South America in comparison with Africa and Asia, with an incidence of 13.4 per 100,000 people per year [2,3,4]. The causes of PD remain unknown, but some studies have attributed it to environmental exposure factors and genetic factors [5]. Moreover, sex and ethnicity have been shown to be influencing factors, with the male:female ratio of patients with PD being approximately 3:2 [3]. Age remains the main risk factor for the development of PD, and the prevalence and incidence of the disease increases exponentially after 60 years of age [3,6]. This trend has important implications for public health, since greater longevity, which is the trend in most countries, is expected to increase the number of people with PD by 50% in 2030 [3,7].

Parkinson disease is associated with non-motor and motor symptoms. Non-motor symptoms include dementia, depression, psychotic characteristics, autonomic dysfunction, oculomotor abnormalities, and olfactory and visual impairments. The motor symptoms include tremor, stiffness, bradykinesia, postural instability, festination, decreased blink frequency, blepharospasm, and Freezing of Gait (FoG) [8]. Medical evaluation of non-motor symptoms is usually performed by a neuropsychologist, while motor symptoms are diagnosed by a neurologist on the basis of medical history, a review of signs and symptoms, and physical and neurological examinations.

The FoG symptom is related on bradykinesia, rigidity, tremor, and postural instability, together with perceptive malfunction and frontal executive dysfunction [8,9]. It presents as a reduction in the forward progression of the feet, despite the person’s intention to walk [10,11,12]. This symptom occurs in 21–27% of patients in the early stages of PD [13,14], and this percentage continues to increase during PD evolution, with the symptom appearing in 80% of patients more than 17 years from the initial diagnosis. FoG may be the cause of falls that can lead to injuries and even death [15].

FoG has its origin in the brain, specifically in the mesencephalic locomotive region (MLR), where the performance of the pedunculopontinal nucleus (PPN) diminishes their connections with basal ganglia. Similarly, the region of the brain stem is related to the condition of freezing, which has been validated with Functional Magnetic Resonance Imaging (FMRI) and ElectroEncephaloGraphy (EEG) data. FoG is no longer considered a strictly motor symptom but is a part of cognitive impairments that originate from areas of the brain that allow the body to be able to walk without hindrance [16,17].

FoG does not respond to existing drugs, and neurorehabilitation exercises tend to be repetitive and tiring. Several invasive methods, such as deep brain stimulation (DBS) [18] or vagus nerve invasive stimulation (VNS) [19], have been developed for the treatment of FoG, but they are expensive, do not guarantee elimination of freezing, and may increase other symptoms. Patients with an episode of FoG can resume walking after receiving external stimulation, and non-invasive methods such as visual, vibratory, and tactile stimulation devices can provide such stimulation at a relatively low cost and without health risks.

Thus, the objective of this study was to analyze non-invasive monitoring systems focused on detection, minimization, and even prediction of FoG, and also evaluate active extracorporeal medical devices that can break FoG episodes. Most devices integrate both functions: monitoring plus stimulation. We aim to compare their underlying processes, methodologies, and outcomes.

## 2. Brain Activity during a FoG Episode

To understand the parameters to be measured by those systems, it was necessary to analyze the brain motor activity involved in PD during episodes FoG. Brain activity occurs when the brain generates electrical impulses known as action potentials, which travel through neurons. Electrical impulses contain information that travels from neuron to neuron making use of hundreds of thousands of them to get transported and perform a specific function, any alteration provokes a change in their contiguous connections [20]. When the brain generates an impulse to move a muscle, the impulse passes through the basal ganglia that help to smooth muscle movements and coordinate changes in posture, such as the gait.

A statistical parametric mapping analysis applied to healthy subjects during the gait revealed that the following areas were activated in their brain activity: supplementary motor, medial primary sensorimotor, striatum, cerebellar vermis, and visual cortex. These results indicate that the cerebral cortexes that control: motor functions, visual cortex, basal ganglia, and cerebellum, may be involved in the bipedal locomotor activities in humans [21]. When a person has PD, there is a degeneration in the cells of the basal ganglia that causes a decrease production of dopamine and reduces connectivity between nerve cells and muscles [22].

Encephalography was used to understand the bioelectrical connections of those who suffer from PD and present FoG. The Figure 1 shows the electrode arranged system in a 10–20 scheme. This scheme was used to analyze brain activity utilizing electrodes on the hair scalp in FOG patients. The presence of FOG episodes generates different levels of energy in the brain waves of the parietal zone (P4), suggesting that this zone has been deeply affected by the disease. Measurements of P4 and the central zone (Cz) are the features that most contributed to the analysis for detecting FoG transition in PD patients [23,24,25].

When the fronto-parietal zone is affected, there is a decrease in executive functions (including cognitive skills), aggravating the problems in people with FoG, who also have failures in their visuospatial network. Some authors such as Amboni et al. compared the progression of cognitive impairment in 26 Parkinsonian patients with FoG (FoG+) and without FoG (FoG-) over a follow-up period of 2 years, finding that FoG+ patients had a faster progression of cognitive impairment, while in FoG- patients, cognitive alteration remained unchanged during this period [26]. Another technique used is magnetic resonance imaging, where it was concluded that FoG+ patients present predominantly frontal-executive dysfunction compared to FoG- patients [27]. At the same time, the right hemisphere of the brain looked more affected in FoG+ patients, which would make sense due to the great influence of this hemisphere on visuospatial abilities [28].

At a neuronal level, the important role of the pedunculopontine nucleus (PPN) located in the MLR, likely play a crucial role in the appearance of axial symptoms in PD. The aim was to activate the remaining PPN cholinergic neurons to improve axial symptoms including FoG and balance deficits in PD patients. Its importance resides in that it is the main center of the mesencephalic locomotor region and controls the initiation, maintenance, and modulation of posture and gait [29]. The presence of FoG was associated with altered functional connectivity between the PPN and the corticopontine-cerebellar pathways (in the bilateral cerebellum and in the pons) and visual temporal areas compared to healthy subjects, additionally marked abnormalities in white matter extending to motor, sensory and cognitive regions [30].

A significant number of PD patients increasingly rely on visual and auditory signals to control the locomotion [31], which creates a problem because visual impairments have been correlated with gait disturbance, deficits in visual attention, memory, and visuospatial abilities [32,33]. Lenka et al. managed to establish that a reduction in inter-hemispheric connectivity between bilateral parietal operculum, somatosensory cortex, and primary auditory areas are correlated with FoG [34]. The hearing deficiency was tested with the Rey Auditory-Verbal Learning Test (RAVLT) [35]. This shows a decline in vision and hearing in those who have episodes of FoG.

## 3. Motor Characteristics during FoG

Although FoG originates in the brain, it manifests as an irregularity in gait. As PD progresses, this irregularity becomes more frequent and disabling for the patient, which results in longer FoG episodes. These episodes can present as complete akinesia when the patient is not medicated (off state, FoG-). When the patient is medicated (on state, FoG+), the FoG episodes are shorter and rarely become akinetic. FoG+ patients present deficiencies during motor initiation that are not present in FoG- patients or healthy people. Additionally, FoG+ patients are slower to initiate motor activity, and in particular, to respond to the signal from the brain to initiate walking. Moreover, these patients require more time to react to stop walking in comparison with FoG- patients and healthy people [36,37].

In muscular-system analyses, FoG is characterized by co-contraction of agonist and antagonist muscles [10,38]. The patients show changes in the pressure of the foot and the behavior of the distance between the steps, as these become shorter compared to those in a slow walk. The frequency of this type of walk is between 4 and 5 Hz [39,40]. Patients with FoG respond to classic auditory, visual, and tactile stimulations [41]. However, these stimulations cannot effectively induce the muscular system to react to bradykinesia [42], which results in alterations in symmetry, rhythm, and bilateral coordination in the patients’ walk with FoG [43,44].

On the other hand, dysfunction in cognitive networks and interictal gait changes may contribute to Parkinson’s disease patients presenting with episodes of FoG. An analysis of patients performing two activities at the same time is presented in one study [45], and the FoG+ patients showed a shorter stride length and slower speed, except during postural balance, in that study. In contrast, during the turn, both groups (FoG+ and FoG-) showed a slower turning speed in the tests involving double activities compared to the findings for the single-task condition.

Analyses of the FoG to date have focused on the lower extremities. However, some studies have revealed determining characteristics in the upper limbs, specifically in the wrists, which show FoG earlier than the legs and feet [36,46]. These studies revealed that FoG can be detected using wrist motion and machine learning models with an FoG hit rate of 0.9 and specificity between 0.66–0.8. Additionally, the standard deviation, acceleration, and rotation were analyzed, in addition to the power between the frequency ranges of 0 to 1 Hz, 5 to 6 Hz, and in the intervals of 0 to 4 Hz, 5 to 8 Hz, and 9 to 12 Hz [46,47].

Some studies have focused on analyzing single-leg posture using portable inertial sensors and performing statistical calculations. The results of these studies indicate that the acceleration peak of the medial-lateral trunk in FoG+ and FoG- patients was significantly lower than that of healthy patients (*p* < 0.05), and the equilibrium was longer in FoG- patients in comparison with that in FoG+ patients [48].

Various studies have revealed significant differences in the duration of the balanced phase of the feet during walking in FoG+ and FoG- patients and healthy subjects. This reduction in the duration of equilibrium is generally associated with an increased risk of falls [49,50,51,52,53].

In one study [54], the two main reasons for falls in PD patients were identified as FoG and impaired balance. This argument is supported by other investigations [55,56,57], which validated the relationship between FoG and falls. These falls are consequences of FoG and appear from the early stages of the disease [58,59,60]. Additionally, falls could be disabling and can deteriorate the quality of life of patients [61,62,63,64]. These falls can occur in multiple directions and are associated with motor symptoms but also with non-motor symptoms, such as mood and cognitive disorders [65]. When the FoG occurs, the center of gravity continues to advance due to the effect of inertia, but the feet stop moving body while the body continues forward, which causes the majority of falls. Therefore, to initiate the walk, FoG+ patients require wider movements than FoG- patients [17].

## 4. Devices Developed for Detection and Stimulation of FoG Episodes

In recent years, various invasive and non-invasive devices have been developed to monitor and thereby minimize FoG. Invasive systems are effective in improving and minimizing some of the symptoms of Parkinson’s disease, but they involve expensive surgical procedures and may be risky for the patient’s health. At present, invasive devices are used when the patients do not respond to drug treatment. The most frequently used invasive systems are those involving deep brain stimulation, stimulation of the vagus nerve, and electrical stimulation of the spinal cord.

Brain stimulation involves three components: an implantable pulse generator (IPG) that houses the battery and electronic components, an implanted electrode, and a lead extender that connects each electrode to the IPG [66]. The electrode is inserted through a small opening in the skull and implanted in the brain, and its tip is positioned within the target area of the brain, which is usually the thalamus, subthalamic nucleus, or globus pallidus. After completion of the procedure, impulses are sent from the neurostimulator to the extension cord and the electrode within the brain; these impulses interfere with and block the electrical signals that cause the symptoms of PD. Deep brain stimulation has been proven to be effective in the management of the symptoms of PD patients but is limited by its complexity and has been associated with mortality rates ranging from 1% to 2% [18].Stimulation of the vagus nerve synapse monoamine optimizes the addition activates the cholinergic neural network circuit that is affected in patients with PD and unlocks up gait. This method activates neurons through stimulation of afferent fibers of the left vagus nerve by a small electrical pulse generator implanted in the upper thorax. The adverse effects, in addition to the risks of implant surgery, are vocal changes and hoarseness [19].Researchers at Duke University have also proposed a new invasive technique based on electrical stimulation of the spinal cord. This procedure involves placing a plate with 16 electrodes on the spinal cord, which stimulates electrical impulses to the neurons to improve the information carried from the legs to the brain so that the patient can regain control of the lower limbs [67].

To minimize the adverse effects of invasive techniques, non-invasive devices that aim to reduce the symptoms of PD, especially FoG, have been developed. Different methodologies have been used for the construction of devices, such as those focused on acquiring EEG signals [24,68,69,70]. Moreover, using accelerometers located at different parts of the lower extremities, including the calf, hip, legs, and ankle [68,71,72,73,74,75], these devices can stimulate the muscle or nerve externally and help patients out of the episode of freezing. All of these devices are aimed at improving the patients’ lifestyles.

Some studies have used video recordings of the progress of PD patients [71,72,76,77]. In one study [78], the authors used a genetic algorithm and an SVM classifier. In another study [79], researchers developed a real-time FoG detection algorithm based on a generalized spectrogram with asymmetric windows, while the simulations in this study show that the proposed algorithm has greater precision than other methods for the detection of the FoG. In one study, an algorithm based on time characteristics and frequency with only a 1-s delay for the identification of FoG was proposed [80]. On the other hand, in another study, a previously trained artificial neural network (ANN) was used [76]. Other researchers have focused on data processing via a discrete wavelet transform-based algorithm that detects FoG episodes in real-time [81]. These systems use an acceleration sensor for data capture.

Some systems used more than one sensor to capture more gait characteristics [80,82,83,84]. The use of inertial sensors (accelerometers, gyroscopes, etc.) provides versatility and good range results in gait analysis. Moreover, the use of pressure sensors contributed to determining the gait phases; however, such sensors cannot accurately determine the distance [85].

Another approach involves calculation of the energy threshold, as described in one previous study [83]; this approach requires 2 s to detect the presence of the FoG and in another study [80], it takes only 1 s. Both approaches compute the power spectral density (PSD).

The increasing penetration of smartphones in the daily life of PD patients has allowed the use of smartphone components, such as the accelerometer, and the data-processing capacity of smartphones for the detection of FoG [86,87]. The use of smartphones to minimize the number of connected external devices for FoG patient monitoring has been presented in a previous study [75,81,87,88].

The existing research on this topic reflects the need for identification of devices for acquiring and processing data as well as for providing FoG-unlocking stimuli [70,88]. The stimuli used can be visual, vibrational, or electrical. In visual stimulation, for example, a projection of parallel lines is used on the floor, which significantly reduces FoG episodes if it is used as continuous stimulation in approximately 51% of the patients and as noncontinuous stimulation in approximately 69% of the patients [81,84,89]. Vibratory stimulation can reduce the time required for resumption of gait and improve the postural relationship of oscillation after only one week of treatment [85]. This form of stimulation is widely accepted by patients because it is comfortable, and women respond faster to a vibratory stimulus when it is placed near the posterior tibial nerve, with a sensitivity of 89% and effectiveness of 96% [74]. On the other hand, electrical stimulation reduced walking time by 19% and FoG episodes by up to 58% [90,91]. In one study [75], the effectiveness of the stimulus for restarting gait was 82.35%, consolidating it as an appropriate method for unlocking the FoG.

A previous study [88] presented research describing the perception of the patient to auditory and vibratory stimuli in an environment with external disturbances. The authors also used glasses and earplugs during testing, simulating the conditions experienced by elderly PD patients. The patients determined that earmuffs are easier and more pleasant to use than glasses. When both were used simultaneously, 66% of the patients perceived glasses as more restrictive than earmuffs. Moreover, in the presence of high levels of external disturbance, the audible system is considered easier to use and more pleasant than the visual system.

Virtual Reality (VR) is another technology explored by different researchers, usually more focused on physical rehabilitation. VR allows you to create personalized rehabilitation programs based on the characteristics of the patient and the progress of the disease, these programs are a highly immersive experience. VR typically uses a lens-type display placed on the patient’s head, sometimes sound, and controls that can be placed on the hand or leg. These devices can be used at home or in physical therapy laboratories, due to their ease of use, relatively low cost, and portability. Specifically, in FoG patients, these devices were used to provide visual stimulus to resume gait [89,92,93].

Table 1 summarizes acquisition approaches used in the studies analyzed in this article, with favorable results in patients with FoG, in Table 2 we did the same based on the used classifier, and in Table 3 based on the used stimulus.

In addition, Figure 2 lists the research on acquisition technologies, processing methods, the transmission means, in relation to the display module and stimulation mechanisms underlying devices aimed at treating the FoG, along with their location relative to the patient.

Figure 2 related references:

Acquisition

Acceleration Sensor [36,68,71,72,73,74,75,79,80,81,83,85,86,87,88,89,92,93,94,95]

Force Sensor [76,85,88,96]

Video Recording [37,71,72,76,77,92,96]

Magnetic Resonance [17,27,30,34]

ElectroEncephaloGraphy (EEG) [24,68,69,70]

ElectroMyoGraphy (EMG) [80]

Location

Head [17,70,73,89,92]

Hip [71,72]

Thighs [80,83]

Achilles Tendon [88]

Ankle [46,82,87,94]

Toe [68,72,76,95]

Foot [73,76,90,92,93,95,96]

Posterior Trunk [86]

Sural Nerve [74,75,81]

Leg [36,37,71,73,74,86,89]

Knees [73,90,97]

Forearms [80]

Shin [73,80]

Shanks [83]

Wrist [36,46]

Chest [36]

Processing

Convolutional Neural Network (CNN) [76]

Short-Time Fourier Transform (STFT) [79,95]

Fast Fourier Transform (FFT) [71,72,94,96]

Discrete Wavelet Transform (DTW) [70,75,87]

Neural Network [80]

Power Spectral Density (PSD) [70,71,80,83,86]

Wavelet [24,68,81]

Statistical methods [36,46,64,65,69,73,75,82,88,89,90,91,92,93,95]

Freezing Index [68,73,79,97]

Hardware

Arduino [74,75,81,85,87,94,97]

Smartphones [46,74,75,76,81,87]

Computer [64,71,79]

Microcontroller [71,76,81,83,88]

Transmission

Bluetooth [36,46,74,75,81,87,94,97]

Radio Frequency [74,76,83,86,87]

WiFi [71,82]

Infrared [73]

Cable Network [83,86,88]

Xbee [85]

Stimulation

Visual [82,84,88,89,92,93,98]

Auditive [70,84,85]

Vibratory [74,75,81,85,88]

Electrical [90,91]

Site

Projection of light on the floor [82,84,89,92,98]

Foot [90]

Tibial Nerve [74,75,81]

Achilles Tendon [88]

Vagus nerve [19]

Dorsal Columns [67]

Chest [82]

Visualization

Smartphone [74,75,81,87,88]

Monitor [65,73,79,82,83,86,90,95,96] 

Led Indicator [71,84]

## 5. Discussion

Since the 1970s, proposals and prototypes have been developed to help understand, detect, avoid, and predict episodes of FoG in patients with PD. Currently, signal-acquisition and processing techniques are still being improved, including analyses of brain activity to understand the origin of the pathology and its connections with motor function. These studies focused on achieving greater sensitivity and specificity in their results aimed at user autonomy, wherein the use of acceleration sensors yielded up to 85% effectiveness in detecting FoG. Similarly, the application of stimuli, such as vibration, addressed FoG episodes by up to 90%, with these results being dependent on the studied population. Research related to FoG episodes has primarily focused on two methods: invasive and non-invasive.

The application of invasive devices has been shown to be effective in minimizing some of the symptoms of PD; however, these devices are associated with numerous risks because the patient must undergo surgery, which requires many hours in the operating room, and is costly. Both factors limited the use of these devices for many patients. Invasive methods are currently being applied when patients do not respond to drug treatment or when the patient wishes to minimize the use of medications. On the basis of the scientific literature, the most frequently applied invasive methods are as follows: deep brain stimulation, stimulation of the vagus nerve, and electrical stimulation of the spinal cord.

With respect to non-invasive devices, the designs primarily involve the use of Inertial Measurement Units (IMU), using accelerometers, for data acquisition in the lower body. Although, the evidence indicates that the use of EEG sensors provide greater signal accuracy and allow faster detection of the presence of FoG in comparison with the inertial sensor response, it is less used because of its higher complexity.

The data is normally transmitted by wireless devices because this technique allows reduction of cost, better ergonomics, and easy implementation. From analyzed papers, Bluetooth technology was the most common choice due to its speed, convenience, and low power consumption. However, since the Bluetooth protocol uses point-to-point connections, the use of this protocol limits the number of concurrent sensors in the system. Other authors have used WiFi networks that allow them to access a server (usually with an Internet link) where record their measurements, and use multiple sensors concurrently.

With advancements in technologies, new hardware prototypes and software algorithms have been appearing, yielding systems with smaller physical sizes and greater processing speeds. This includes the use of greater sampling frequencies ranging from 10 Hz to 1 KHz for processing, allowing the application of frequency analysis techniques (such as FFT and DWT) or the use of neural networks. However, a higher sample rate analysis does not guarantee better results. It must be in accordance with the activity to be measured, since motor responses would hardly reach high frequencies, contrary to electroencephalographic signals. The processing system also needs to be at the level of data generation and complexity, greater demand for systems that use electroencephalographic signals and less in the case of IMU.

Although most studies and systems are focused on the characterization of FoG, some of them also made the next step and implement solutions to unblocking detected FoG episodes. Since the symptoms precede the cognitive function decline that underlies the pedunculopontine nucleus, methods have been proposed to reactivate this neuronal activity through visual, auditory, haptic, or electrical stimulation. Their functions are based on the patient concentrating on an external signal so that they can redirect their attention from freezing and the neuronal link can be reactivated to allow them to resume their walk.

Some authors recommend in future studies, that patterns in PD patients’ gait will be determined to allow prediction of FoG episodes. In addition, exists a consensus about the importance of database analysis of walking data is carried out using artificial intelligence to determine the possible causes of FoG.

We detected that few studies have focused on seeking standardization of algorithms to determine FoG and that these do not depend on posture or location of sensors, which would mean giving patients independence.

## 6. Conclusions

FoG is one of the most disabling consequences of Parkinson’s disease. Many studies have explored different approaches for its detection and overcoming. In this review, we have made a classification and compendium of all of them. Synthesized in Table 1, Table 2 and Table 3 and Figure 2.

As has been indicated in the discussion, the advancement of technology is allowing the improvement of the effectiveness of the techniques already proven, at the same time that the usability of the devices is improved. In any case, the study of devices for detecting and overcoming FoG is still a field open to new techniques.

## Figures and Tables

**Figure 1 healthcare-10-00976-f001:**
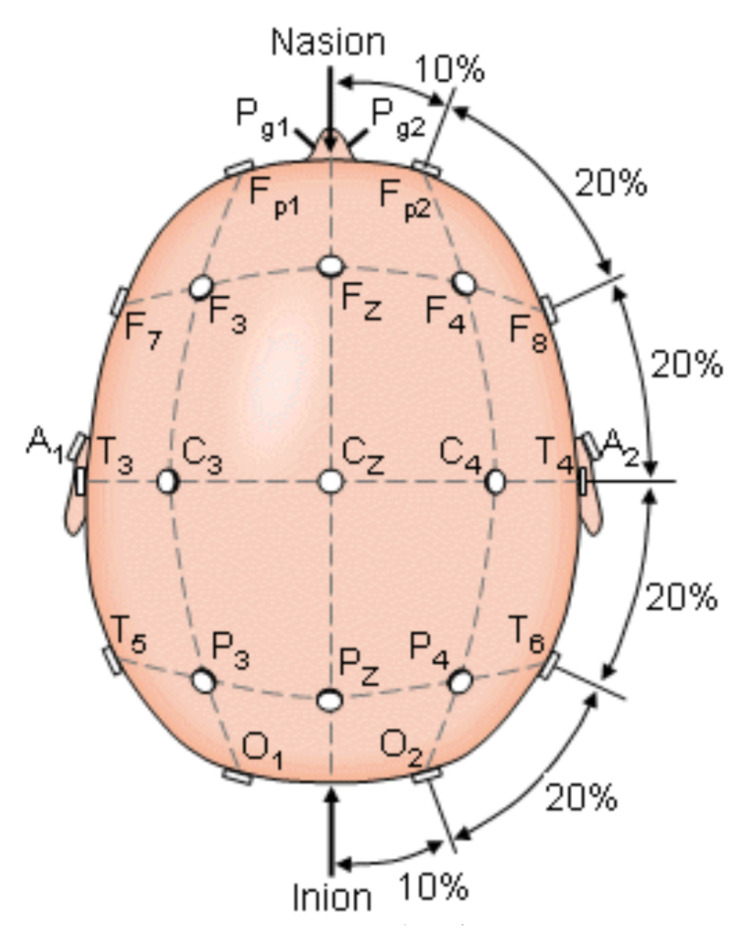
The international 10–20 system seen above the head. A = Ear lobe, C = central, Pg = nasopharyngeal, P = parietal, F = frontal, Fp = frontal polar, O = occipital [23].

**Figure 2 healthcare-10-00976-f002:**
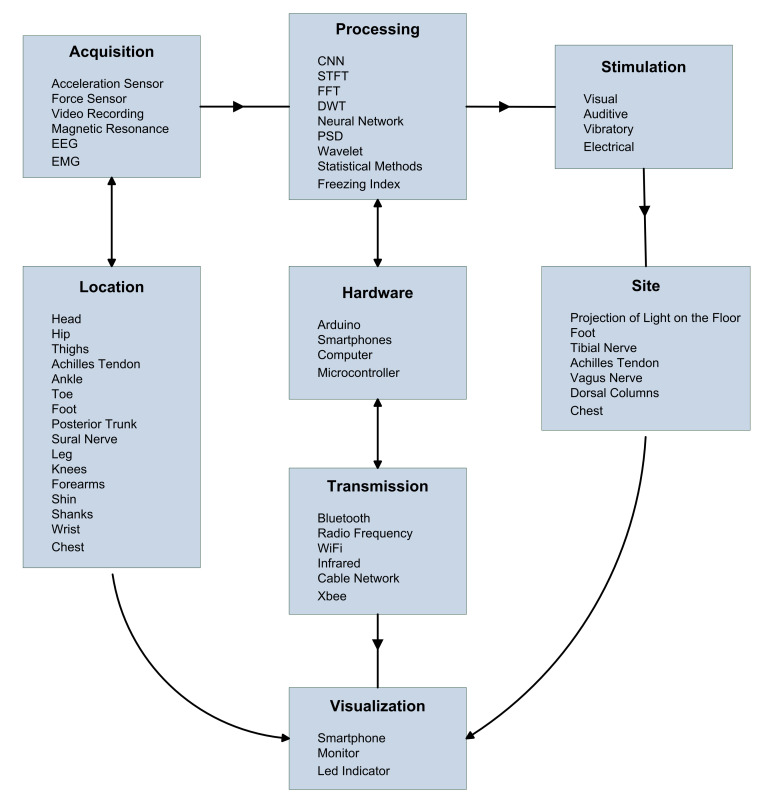
Analysis of non-invasive techniques most used in patients with FoG: acquisition systems, devices, transmission, visualization, data processing, and stimulation. ElectroEncephaloGraphy (EEG), ElectroMyoGraphy (EMG), Convolutional Neural Network (CNN), Short-Time Fourier Transform (STFT), Fast Fourier Transform (FFT), Discrete Wavelet Transform (DTW), and Power Spectral Density (PSD).

**Table 1 healthcare-10-00976-t001:** Description of devices developed until today to help patients with episodes of FoG based in the acquisition type.

Acquisition	Description	Results and Efficiency
Acceleration sensor [71,80,81,83,94,95,97]	Tri-axial Sensors, which extract the features of the changes of acceleration in the march.	The use of sensors determines the motor state. They show small intensity on having been standing up and before a FoG episode.
Force Sensor [76,85,88,96]	They receive the pressure in the areas of the plant of the foot.	Improvement in the time peak of pressure of the heel, the moment peak of pressure of the toe, the time in the sensor of heel and the position of oscillation after the treatment.
EEG [24,69,70]	System of 4 electrodes located in 4 areas of the skull that receive bioelectric stimuli.	Greater speed in the detection of FoG. Only the difference of the channels O1-T4 and P4-T3 give information about the FoG.
Video recordings [71,72,76,77,88]	Recorded walks, where an expert determines the presence of FoG and analyzes the motor posture.	Recordings under TUG (Timed Up And Go) test. Combined with other acquisition methods and used as a test.

**Table 2 healthcare-10-00976-t002:** Description of devices developed until today to help patients with episodes of FoG based in the used classifier.

Classifier	Description	Results and Efficiency
Artificial neural network [24,68,76,78,79,80,81,96]	Multilayer perceptron.	The sensitivity, specificity and precision in the analysis it was 82%, 77% and 78% respectively.
	Diffuse logic.	Greater capacity to reduce the detection of false negatives, sensitivity of 89% and a specificity of 97%.
	Backpropagation.	Average precision values, sensitivity and specificity are around 75%
Thresholds [72,73,74,79,95,97]	Freezing indexes (FI).	Commonly combined with energy levels for detection. Low levels of FI before the FoG.
	Inertial measurements taken from normal march.	Rules for the entry of normal behavior patterns
Algorithm [69,70,71,72,75,81,86,94]	FFT for analysis of time series, often it combined with other statistical analyzes.	The frequency band of the FoG (3 to 8 Hz), also its duration and the number of episodes was determined.
	DWT as localized energy analysis.	Sometimes it has misalignment results. It allows comparing similar patterns instead of just a specific pattern in the time subsequence.
	PSD as is the measure of signal’s power content versus frequency.	The PSD was calculated for each trial using a 4-s Hanning window with 50% overlap.

**Table 3 healthcare-10-00976-t003:** Description of devices developed until today to help patients with episodes of FoG based in the used stimulus.

Stimulus	Description	Results and Efficiency
Visual [82,84,98]	Projection of parallel lines spaced apart on the ground, perpendicular to the view of the patient.	Further reduces the freezing medium (69%) on request signals but reduces the number of FoG (43%) in continuous signals.
Auditive [70,84,85]	Emission of an audible signal by a handset in the presence of the FoG.	In the presence of a double disruption (visual and auditory), the audio system is easier to use and more pleasing to the sensory.
Vibratory [74,75,81,85,88]	Micromotors located at the lower end for vibrotactile stimulation.	A tactile sensory system is able to impose a rate despite sensory disorders was demonstrated.
Electric [90,91]	Electric shock directed to a muscle, to produce a controlled shrinkage.	Decreased to 19 % walking time and FoG episodes were reduced up to 58%.

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
