# Peer review of "Review of Active Extracorporeal Medical Devices to Counteract Freezing of Gait in Patients with Parkinson Disease"

_healthcare, 2022, doi:10.3390/healthcare10060976_

Round 1

Reviewer 1 Report

The article presents an interesting topic. However, it needs some corrections and additions.

  • Please unify the FoG/FOG abbreviation.
  • Explanations of the abbreviations used in Fig. 2 are missing.
  • Table 1 is very unreadable. Please reorganise it in a more reader-friendly way. I suggest that the division in the table should refer to figure 2 somehow. 
  • Please check the citation order - it is not always followed. 

Author Response

Comment 1:  The article presents an interesting topic. However, it needs some corrections and additions.

Thank you for the positive comments

Comment 2:  Please unify the FoG/FOG abbreviation.

Mistyping of FOG / FoG fixed, unified to FoG

Comment 3:  Explanations of the abbreviations used in Fig. 2 are missing.

Added in the figure caption an explanation for each abbreviation:

ElectroEncephaloGraphy (EEG), ElectroMyoGraphy (EMG), Convolutional Neural Network (CNN), Short-Time Fourier Transform (STFT), Fast Fourier Transform (FFT), Discrete Wavelet Transform (DTW), and Power Spectral Density (PSD)

Comment 4:  Table 1 is very unreadable. Please reorganise it in a more reader-friendly way. I suggest that the division in the table should refer to figure 2 somehow. 

Table 1 has been divided into 3 different tables to improve understanding

 Table 1 summarizes acquisition approaches used in the studies analyzed in this article, with favorable results in patients with FoG, in table 2 we did the same based on the used classifier, and in table 3 based on the used stimulus.

Comment 5:  Please check the citation order - it is not always followed. 

Citation order was checked and corrected

Reviewer 2 Report

This present study surveys the state of the art in applications of invasive and non-invasive devices to help understand the origin of the pathology of FoG episodes and its connections with motor function in patients with Parkinson’s disease, and discusses the use of related technologies aimed at treating the FoG. The authors have done an exhaustive work in pursuing the study and digging out previous researches in this field. The manuscript is well-written and can be considered for publication if authors revise based on following suggestions:

  • The abstract seems to be a little misguiding from the content in the main manuscript. It can be rewritten to make the readers understand better.
  • The introduction seems too long and includes unnecessary statements which are not required. It is better if point to point statement is made related to the objective of the review. E.g., line 40-42 “The systems mentioned in previous studies [9–12] have contributed to quantitative” seems that this sentence is not related to previous one or may be rephrasing is required.
  • Line 96 Amboni and collaborators instead use Amboni et. al., and also Line 118 Lenka et. al.,. 
  • The paragraph on section 3 seems to be hard  to understated or need more elaboration. It is better if the authors elaborate more on the symmetry, rhythmicity, and bilateral coordination, which are the main symptoms accompanied with FoG. Moreover, 
  • The discussion seems to be too concise with the inability to summarize the authors’ point of view. The topics are being cherry-picked and this review seems to be more of a narrative review
  • The non-invasive techniques for detecting FoG are not properly described. Also, it is better if studies are summarised based on disorders since it would be easy for the reader to write
  • It would be nice if authors included Virtual Reality as a promising non-invasive method and the technology quite mature to be used in gait research. I would suggest looking at the following references that could improve sections of the manuscript along with other non-invasive devices:
  • 1. Janeh, Omar, and Frank Steinicke. "A Review of the Potential of Virtual Walking Techniques for Gait Rehabilitation." Frontiers in Human Neuroscience 15 (2021).
  • 2. Keshner, Emily A., and Anouk Lamontagne. "The untapped potential of virtual reality in rehabilitation of balance and gait in neurological disorders." Frontiers in virtual reality 2 (2021): 641650.
  • 3. Canning, Colleen G., et al. "Virtual reality in research and rehabilitation of gait and balance in Parkinson disease." Nature Reviews Neurology 16.8 (2020): 409-425.
  • Authors mentioned that placing lines on the floor used as rehabilitation technique for FoG and in this context I would suggest the following references: 
  • 1. Gómez-Jordana, Luis I., et al. "Virtual footprints can improve walking performance in people with Parkinson's disease." Frontiers in Neurology (2018): 681.
  • 2. Janeh, Omar, et al. "Gait training in virtual reality: short-term effects of different virtual manipulation techniques in Parkinson’s disease." Cells 8.5 (2019): 419.
  • The study is missing conclusion . Suggest authors to conclude their work.
  • The whole manuscript lacks vision and it is suggested for the authors to clearly summarize what message the readers will draw post reading this manuscript. 

Author Response

Comment 1:  This present study surveys the state of the art in applications of invasive and non-invasive devices to help understand the origin of the pathology of FoG episodes and its connections with motor function in patients with Parkinson’s disease, and discusses the use of related technologies aimed at treating the FoG. The authors have done an exhaustive work in pursuing the study and digging out previous researches in this field. The manuscript is well-written and can be considered for publication if authors revise based on following suggestions:

Thank you for the positive comments

Comment 2:  The abstract seems to be a little misguiding from the content in the main manuscript. It can be rewritten to make the readers understand better.

The abstract was rewritten:

Parkinson Disease (PD) primarily affects older adults. It's the second-most common neurodegenerative disease after Alzheimer’s disease. Currently, more than 10 million people suffer from PD, and this number is expected to grow, considering the increasing global longevity. Freezing of Gait (FoG) is a symptom present in approximately 80% of advanced-stage PD's patients. FoG episodes alter the continuity of gait, may be the cause of falls that can lead to injuries and even death. The recent advances in the development of hardware and software systems for monitoring, stimulus, or rehabilitation of patients with FoG has been of great interest to researchers because detection and minimization of the duration of FoG events is an important factor in improving the quality of life. This article presents a review of the research on non-invasive medical devices for FoG, focusing on the acquisition, processing, and stimulation approaches used.

Comment 3:  The introduction seems too long and includes unnecessary statements which are not required. It is better if point to point statement is made related to the objective of the review. E.g., line 40-42 “The systems mentioned in previous studies [9–12] have contributed to quantitative” seems that this sentence is not related to previous one or may be rephrasing is required.

Lines 40 to 42 were cut, they were part of a longer paragraph deleted during the last version for MPDI, with this version they are unnecessary. Also, some minor changes were made to improve readability in the rest of the paragraphs.

Comment 4:  Line 96 Amboni and collaborators instead use Amboni et. al., and also Line 118 Lenka et. al.,. 

Thanks, we corrected both.

Comment 5:  The paragraph on section 3 seems to be hard to understated or need more elaboration. It is better if the authors elaborate more on the symmetry, rhythmicity, and bilateral coordination, which are the main symptoms accompanied with FoG. Moreover, 

 Our work is focused in FoG and not in its collateral symptoms, for that reason, we didn't include an explanation of them. Section 3 was rewritten to improve understanding.

Comment 6:  The discussion seems to be too concise with the inability to summarize the authors’ point of view. The topics are being cherry-picked and this review seems to be more of a narrative review.

We summarized different authors' approaches to FoG monitoring, FoG detection, and walk resuming. We detected only the untreated topic of FoG detection systems standardization.

Comment 7:  The non-invasive techniques for detecting FoG are not properly described. Also, it is better if studies are summarised based on disorders since it would be easy for the reader to write

From line 195 to 263 the detection techniques are described and are summarized in Table 1.

 Comment 8:  It would be nice if authors included Virtual Reality as a promising non-invasive method and the technology quite mature to be used in gait research. I would suggest looking at the following references that could improve sections of the manuscript along with other non-invasive devices:

  • Janeh, Omar, and Frank Steinicke. "A Review of the Potential of Virtual Walking Techniques for Gait Rehabilitation." Frontiers in Human Neuroscience15 (2021).
  • Keshner, Emily A., and Anouk Lamontagne. "The untapped potential of virtual reality in rehabilitation of balance and gait in neurological disorders." Frontiers in virtual reality2 (2021): 641650.
  • Canning, Colleen G., et al. "Virtual reality in research and rehabilitation of gait and balance in Parkinson disease." Nature Reviews Neurology16.8 (2020): 409-425.

We read suggested papers, and some of promising papers from their references. Most of them are about VR but not specifically focused in FoG. We included a new paragraph about VR, and selected references are also include in tables and figure 2.

Virtual Reality (VR) is another technology explored by different researchers, usually more focused on physical rehabilitation. VR allows you to create personalized rehabilitation programs based on the characteristics of the patient and the progress of the disease, these programs are a highly immersive experience. VR typically uses a lens-type display placed on the patient's head, sometimes sound, and controls that can be placed on the hand or leg. These devices can be used at home or in physical therapy laboratories, due to their ease of use, relatively low cost, and portability. Specifically, in FoG patients, these devices were used to provide visual stimulus to resume gait

Comment 9:  Authors mentioned that placing lines on the floor used as rehabilitation technique for FoG and in this context I would suggest the following references: 

  • Gómez-Jordana, Luis I., et al. "Virtual footprints can improve walking performance in people with Parkinson's disease." Frontiers in Neurology(2018): 681.
  • Janeh, Omar, et al. "Gait training in virtual reality: short-term effects of different virtual manipulation techniques in Parkinson’s disease." Cells8.5 (2019): 419.

Good references we added both reference in our text, tables and figure 2. Also, add a new reference:

 Georgiades, M. Gilat, K. Ehgoetz Martens, C. Walton, P. Bissett, J. Shine and S. Lewis, "Investigating motor initiation and inhibition deficits in patients with Parkinson’s disease and freezing of gait using a virtual reality paradigm", Neuroscience, vol. 337, pp. 153-162, 2016.

Comment 10:  The study is missing conclusion .Suggest authors to conclude their work.

Conclusions section was added

Conclusion

FoG is one of the most disabling consequences of Parkinson’s disease. Many studies have explored different approaches for its detection and overcoming. In this systematic review, we have made a classification and compendium of all of them. Synthesized in tables 1, 2, and 3 and Figure 2.

As has been indicated in the discussion, the advancement of technology is allowing the improvement of the effectiveness of the techniques already proven, at the same time that the usability of the devices is improved. In any case, the study of devices for detecting and overcoming FoG is still a field open to new techniques.

Round 2

Reviewer 2 Report

Thank you for responding to my comments. Your answers are clear and your changes to the manuscript seem appropriate to me. However, the current revised version of this research is well written and well structured. Therefore, I endorse the publication of this manuscript in its current form.